# The Gaitprint: Identifying Individuals by Their Running Style

**DOI:** 10.3390/s20143810

**Published:** 2020-07-08

**Authors:** Christian Weich, Manfred M. Vieten

**Affiliations:** Sports Science, University of Konstanz, 78464 Konstanz, Germany; manfred.vieten@uni-konstanz.de

**Keywords:** attractor method, human cyclic motion, running quality, individual locomotion, recognition

## Abstract

Recognizing the characteristics of a well-developed running style is a central issue in athletic sub-disciplines. The development of portable micro-electro-mechanical-system (MEMS) sensors within the last decades has made it possible to accurately quantify movements. This paper introduces an analysis method, based on limit-cycle attractors, to identify subjects by their specific running style. The movement data of 30 athletes were collected over 20 min. in three running sessions to create an individual gaitprint. A recognition algorithm was applied to identify each single individual as compared to other participants. The analyses resulted in a detection rate of 99% with a false identification probability of 0.28%, which demonstrates a very sensitive method for the recognition of athletes based solely on their running style. Further, it can be seen that these differentiations can be described as individual modifications of a general running pattern inherent in all participants. These findings open new perspectives for the assessment of running style, motion in general, and a person’s identification, in, for example, the growing e-sports movement.

## 1. Introduction

Quantitatively describing and understanding the characteristics of a well-developed and efficient running style is a central issue in athletic sub-disciplines. It is not without reason that running efficiency is considered to be one of the three determinants of endurance performance alongside aerobic capacity (VO_2max_) and the fractional utilization of the VO_2max_ [1,2]. The identification of subject-specific running characteristics is crucial for approximating towards a better understanding of running efficiency from a biomechanical standpoint. Since the early seventies, experimental psychology has demonstrated high identification rates of so-called bio-motion animations [3]. Here, viewers were able to recognize gender, body physique, tension, and even the mood of a walking person simply by watching white dots on a black screen representing their major joints. While subjective observation data [4,5], the understanding and interpretation of human motion recognition, is certainly not a new endeavor, smart surveillance, robotics, medical applications and others, as well sports and exercise have taken advantage of the growing technology and methods ([6], Table 1) within the last two decades. Especially, the analyses of athletic movements and postures may potentially improve the athlete´s competitiveness and reduce the injury risk. Furthermore, within the last decade, with the field of e-sports another interesting area of application has emerged. Since 2006, when Nintendo released its Wii console, motion-controlled gaming systems have evolved so much that they are now used for therapeutic [7] and educational purposes [8]. Today, there are two application methods when dealing with human motion recognition: the vision- or video-based approach and the use of wearable sensor technology worn by the user. Before the latter was introduced to the mass market in the past decade, earlier work, like that by Schoellhorn and Nigg [9], relied on force plates and mainly video data as an effective tool to identify locomotion characteristics. In the late nineties, [10] provided a concise article about what was possible in terms of human motion analysis at that time. The recognition of persons or activities on the basis of body parts was seen as fundamental, mostly in comparison with pre-defined models [3] without separating into body parts, but rather focusing on the person or activities as a whole based on consecutive image sequences. Later, they revised their earlier work, which concentrated on simple actions, and expanded it with methodologies dealing with human actions, interactions and group activities [11]. Until the end of the 2010s, studies on the recognition of persons based on their gait were largely based on these technical circumstances. For example, Goffredo et al. [12] presented. for the first time, a methodology in which video sequences with widely different walking persons achieved a recognition rate of 92.2%. The recognition was marker-free and without prior calibration of the cameras. In 2011, Lin [13] focused in their work on kinematic and kinetic parameters of the lower limb joints. The group combined a marker-based motion capture system with force plates with a self-organizing neural network map on the software side. Their biometric methodology, which did not use a marker-free approach for reasons of precision, achieved a very high recognition rate of 99.07% on average for all examined joints (hip, knee, ankle).

In contrast to these traditional raw data, as well as data from optical motion capture or magnetic tracking systems, modern research rather draws upon inertial sensors collecting acceleration and gyrometer data. The latter devices are constructed as a micro-electro-mechanical-system (MEMS) and are useful due to with their size, weight, cost-efficiency, and low power consumption [14,15]. They have also been shown to be convenient and easy-to-use in different practical sport settings and rehabilitation related contexts [16,17]. A recent study [18] in the sport of basketball focused on the use of MEMS sensors to detect complex, sports-typical movements of the upper extremities, such as dribbling, catching, passing, and throwing. The group used well-known computer-based evaluation methods (Principal component analysis and support vector machine) and presented a highly reliable method with an average recognition rate of 96%.

MEMS sensors collect joint coordinate tracks, which can be compressed into an entity called an attractor. An individual attractor can be regarded as a mean cycle derived from multiple gait or running cycles during a session, lasting from seconds up to several minutes [19]. Based on the latter, current works in sports rehabilitation propose a highly individual foot acceleration print [20] or gaitprint [21] representing a unique characteristic of a person´s gait. They can be understood as an analogue to a fingerprint, representing a singular pattern of a human finger, which will only match with one particular person. Assigning a gaitprint to single individuals would open the possibility to identify people based only on their stride characteristics.

The Attractor Method [19] uses acceleration and gyrometer raw data derived from MEMS sensors to produce sensitive results allowing for the objective analysis of subtle attributes in movement patterns and its variation. Vieten et al. [19] and others [21] reported that human cyclic motions, such as running or cycling, can be described as limit-cycle attractors and thus be used to create individual gaitprints from whole movement sequences. Individual attractors, created by a computer algorithm, representing the full motion cycle data of the moving subject, can be stored in a database for further analysis. Once a new short running sequence is recorded, at a later time it can be compared with the database to screen out the dataset with the highest concordance. In a theoretical paper, Vieten and Weich [22] demonstrated that this process enabled discrimination between an individual’s and other persons’ running patterns. Moreover, the authors described extra components of human cyclic motion—attractor morphing, short time fluctuations, transient effect, control mechanism, and technical noise—adding further variation to the individual limit-cycle attractor, which must be taken into account during the recognition process [22] (p. 3).

The aim of the current study was to establish that athletes can be recognized by their running style. It is shown that an algorithm, which is based on the Attractor Method, is capable of recognizing a running person based solely on data received from MEMS sensors. The present study demonstrates that the detection rate is as high as 99%, while the false identification probability is 0.28% overall. From this, it can be concluded that running style is highly individual. Beyond the recognition capability, the Attractor Method approach further aimed to highlight the kinematics of this procedure and to pave the way for a better understanding concerning running quality in athletic contexts.

## 2. Materials and Methods

A total of 30 athletes (Table 1), 9 females and 21 males, were tested from April till July 2018 in Kreuzlingen, Switzerland (Nationale Elitesportschule Thurgau). All of the participants were active and experienced runners, none did show any present injury signs at this point, which could have possibly impeded their performance. The only prerequisites were to be aged 18 years or older and able to run 20 min. without reducing the pace, which had been determined in advance by a lactate threshold test. The participants started with a speed of 8 km/h, which was increased every 3rd minute by 0.5 km/h separated by 30 s pause to take a lactate sample until exhaustion. The study was approved by the local Ethical Committee of the University of Konstanz, Germany, under the RefNo: IRB20KN10-009. All of the participants filled out and signed an informed consent.

To collect the necessary raw accelerometer data, two inertial sensors were used (RehaWatch by Hasomed. Magdeburg, Germany), which were attached to both ankles by a hook-and-loop fastener. The latter assured a stable fixation right above both lateral malleoli, which guaranteed the sensor positioning to be identical in all trials. The sensors have a size of 60 × 35 × 15 mm and weigh 35 g each. They function as triaxial accelerometers with up to 16 g ($1\;g=9.81\;m/s^{2}$), triaxial gyroscopes with up to 2000°/s and a magnetometer, which data that were not used within this study. The sampling rate was consistently set at 500 Hz. The acceleration data were gathered while using the app RehaGait Version 1.3.9 of Hasomed (Magdeburg, Germany) with data saved to a smartphone (Samsung Galaxy J5, Seoul, South Korea). The recordings of the feet were collected in three dimensions (x, y, z) in the coordinate system co-rotating with the legs. The hardware to measure the lactate content of the blood was a Lactate Scout and proper lactate sticks (by Senslab GmbH, Leipzig, Germany). The software Ergonizer (by Prof. Dr. Kai Röcker, Freiburg, Germany) was used to compute the individual anaerobic threshold.

The participating athletes repeated the testing protocol in a timeframe of approximately two weeks consisting of four testing days separated by at least 24 h. On the initial test day, they performed a lactate step-test (LT) to determine their individual anaerobic threshold. Further tests on days two to four (run 1–3 decoded R1–3 for further use) consisted of 20 min. of running on a treadmill. The speed for all sessions was set to a running pace according to 95% of the lactate threshold speed. During all tests, the participants were equipped with two activated acceleration sensors, as described above, attached to the ankles atop each lateral malleolus. The smartphone to collect the data was placed on a desk beside the treadmill to ensure undisturbed reception.

Further analysis required the collected 20-min. data block to be divided into 60 s intervals. A file splitter was applied to produce 20 single datasets. The raw data text-file contained thirteen columns: time and the acceleration as well as the gyroscope data in x, y, and z directions for the left and the right foot, respectively. Afterwards, the Attractor App was used to calculate the attractor data of each one-minute data set. The functionality of the Attractor App is based on the attractor method developed by [19] and that uses acceleration and gyroscope data to produce sensitive results allowing the objective analysis of subtle attributes in movement patterns and its variation. The app is available online via http://www.uni-konstanz.de/FuF/SportWiss/vieten/CyclicMove/. Furthermore, one of the running data sets of each subject (only minutes 11 to 20 of R1, R2, or R3) was taken to calculate a mean attractor, named super attractor, representing the individual running pattern. In the calculation data from the left and the right foot were taken to establish the super attractors of both feet. The advantage of the super attractor, a mean of 10 attractors of one running session, in contrast to a comparison with a single minute, is the avoidance of outlier or extreme attractors. For the latter, calculation only of minutes 11–20 were included, to ensure that the athletes had already left their transient phase [22]. All of the datasets were speed-normalized according to their individual running pace to be comparable. Additionally, to prepare for the recognition analysis, a *recognition horizon* around each single attractor point, was calculated. This horizon was defined as the volume area at a distance equal to five standard deviations around each attractor point. The choice of the data set (R1–R3) chosen for calculating the super attractor was balanced over the three trials. This compensated the influence of a possible bias, such as the learning effect. By creating 30 super attractors (due to 30 subjects) with their associated horizon, a catalogue was generated to undertake the identification analysis. All other runs, two of each subject, constituted a comparison pool.

To identify a person based on the running motion, any dataset, independent of the data contributing to the respective super attractors and containing at least 50–60 cycles, could be taken from any session. The algorithm was applied to all super attractors from the catalogue using a point to point analysis to compare a newly chosen dataset to the super attractor (Figure 1). The outcome of each tested comparison was a similarity rate, which was defined as the percentage of data points lying within the recognition horizon.

In this identification process all one-minute running sequences were compared with all super attractors in the catalogue. Altogether, 18,000 similarity rates (30 subjects × 2 runs × 10 sequences when compared to the 30 right and left side super attractors) were calculated. The comparison of a super attractor with attractors of the last 10 min. of a run, being runs as compared of one single individual or runs of a different individual, constitute a random event. As such, from a theoretical standpoint, the results are distributed normally with a definite mean and standard deviation. In addition, normality was numerically tested. With this, the probability of not identifying a subject can be set, in our case as α=0.01. The border discrimination between different and same persons’ similarity rate can then be calculated with the help of Equation (1).
(1)$$\alpha =\overset{T}{\underset{-\infty }{\int }}\frac{1}{\sqrt{2\pi }\cdot \sigma_{s}}\cdot e^{-\frac{{\left(x-\mu_{s}\right)}^{2}}{2\sigma_{s}^{2}}}dx=\mathrm{erf}\left(\frac{T-\mu_{s}}{\sqrt{2}\cdot \sigma_{s}}\right)$$

This results in
(2)$$T=\mu_{s}+\sqrt{2}\;erf^{-1}\left(2\;\alpha -1\right)\;\sigma_{s}$$
Finally, the probability of a false identification can be calculated while using Equation (3).
(3)$$p=\overset{\infty }{\underset{T}{\int }}\frac{1}{\sqrt{2\pi }\cdot \sigma_{d}}\cdot e^{-\frac{{\left(x-\mu_{d}\right)}^{2}}{2\sigma_{d}^{2}}}dx=1-\mathrm{erf}\left(\frac{T-\mu_{d}}{\sqrt{2}\cdot \sigma_{d}}\right)$$
Here, erf(…) is the error function and ${\mathrm{erf}}^{-1}\left(\ldots \right)$ the inverse. *x* represents the similarity rate, μ and σ, respectively, the mean and standard deviation of the probability density that describes the similarity rate of same subjects with the index s and for different subjects with the index d. A graphic illustration of the relationships is provided in Figure 2.

Based on this general outcome, in the second step, the identification process was repeated for all super attractors to be compared only with the “worst-case data set”, meaning the subject who had the highest false recognition rate. We identified the similarity rates to be normally distributed (Kolmogorov–Smirnov statistic with *p* > 0.05) and calculated the means and standard deviations using Equations (1)–(3).

Recognition of a subject depends on the complete motion description, including all the kinematics. The methodology of [22] was used to determine the influencing factors and the magnitude of attractor variations. They describe the kinematics of human motion as a superposition of six contributing terms:
Limit-cycle-attractor, a closed line in acceleration space, representing the characteristic main contribution, which repeats in each cycle. Attractor morphing, a slow deviation, deforming the actual attractor within well definable borders.The transient effect occurring as temporary oscillations at the onset of a cycling movement, of which the strength decreases rapidly as a negative exponential function depending on time. Such initial transient oscillations can generally be found in many dynamical systems, like human neurology [23] or biomechanics of muscles [24].Short-time fluctuations in the form of a “random walk” around a morphed attractor.The controlling mechanism, which kicks in when current accelerations deviate too much from the morphed attractor.MEMS sensors’ noise


The latter is treated as white noise cancelling out over the course of time (mean = 0). A similar cancellation counts for the controlling mechanism in combination with the short-time fluctuations. These two effects together have a mean contribution of zero due to the random positive and negative contributions in three dimensions.

Super attractors were used to be compared with independent measurements of the single subjects to identify the magnitude of the individual attractor morphing. The outcome is a parameter termed *δM*, as described in a paper introducing the attractor method [19] (p. 3) *δM* expresses the velocity normalized difference between two attractors, containing information about changes in the individual running pattern. In general, the smaller *δM* is, the more similar are the two compared attractors. Thus, it serves as an indicator for the magnitude of the morphing process.

Further, Vieten and Weich [22] reported that each continuous run is potentially accompanied by an initial transient phase lasting approximately 4 to 10 min. Based on this assumption, all of the running data included in the analysis of the gaitprint procedure only contained data that were collected after the initial ten minutes to exclude the impact of the transient effect.

## 3. Results

### 3.1. The Similarity Procedure

The similarity procedure created a distribution of similarity rates presenting the same subject comparisons (green dots for run 1 and black × for run 2) and comparisons of subjects’ super attractor with the worst case run of a different person (gray bars) (Figure 3a,b). For almost all cases, there is a distinct gap visible between the same- and the different subject comparisons. A sign that the recognition of a person is achievable with high probability.

The outcomes based on Equation (3) concerning the probability of detecting a false positive assignment, when checking for an individual running pattern, are, on average, below 1% with a maximal false detection rate of approximately 9% (subject 11). Figure 4 shows a complete overview of all participants.

### 3.2. Morphing and Transient Analyses

In addition to the overall results of the recognition process, the morphing and transient analysis as a precondition are outlined (Figure 5 and Figure 6) below. In general, it can be stated that a low *δM* is connected to a high similarity between the compared attractors.

Figure 5 shows all three *δM* means of each subject separated into their individual boxes. The *δM* mean represents the average of all results comparing a super attractor against the single running from minutes 11 to 20. It is evident that the general magnitude of the morphing process is rather small, lying in a range of *δM* = 7 to 14, and the individual values within single subjects do not vary to a great extent. From earlier studies, it is known that, if a running motion is joined by transient oscillations [22], their impact very likely subsides by ten minutes. The procedure to visualize the transient effect was applied to a series of data sets in order to confirm this observation for the underlying data of the current study. Figure 6 shows a selection of four subjects derived from data taken from the current study with different initial transient oscillations. It can be seen that these oscillations level off before minute ten. The selections provided are severe cases and it should be noted that other subjects showed less or no transient effect. Summarized, all of the characteristic constants, derived from a curve fitting process, as described in [22] and using the curve fitting software (CurveExpertPro, version 2.6.5, Hyams Development), can be seen in Table 2. This overview serves as guide values for future studies. Furthermore, the intervals in the current study can be classified as small.

## 4. Discussion

The hypothesis of the present study stated that a running person can be recognized from stride data derived from MEMS sensors attached to the ankles. The results indicate that the chance of having a false positive allocation is, on average, as small as 0.28%. Also meaningful is the high extent of subjects having the chance of being identified incorrectly is almost zero. Only two subjects were recognized incorrectly, with a probability of 5% or slightly higher.

Consequently, the presumption of a gaitprint, defined as above and proposed in earlier works by [20,21,22], can be highly confirmed. Morphing can deform an attractor in many different ways, which most probably results in *δM*s of comparable values as can be seen in Figure 4. This leads to the assumption that the impact of the morphing on the recognition process is at best marginal. Based on the outcome of [22] and the analysis of the data that were collected for the current study, it was shown (Figure 6), that, although a person’s motion was accompanied by a transient effect at the onset, it subsided by 10 min. at the latest. For this reason, it can be expected that the running data from minute 11 until 20 are not influenced by transient oscillations. Thus, it can be assumed that the recognition analyses in this paper were not affected by the transient effect. In addition, it should be taken into account that the influence of the transient effect could be linked to particular individual characteristics, such as the performance level. For example, Strohrmann et al. [25] considered, among other things, the change in ground contact time within a step cycle. The participants were divided into performance groups based on their training kilometers and running speed. They ran at a running speed of 75–85% VO2max, thus with a pace comparable to the 20 min. in the presented study. There were significant differences in the absolute value of the ground contact time (running beginners > running experts). While the contact time of the experienced runners levelled off after a few minutes, the contact time of the beginners increased over 15 min. until it remained stable until the end. While future investigations must be carried out to gain deeper information about the described transient process, the performance level data from the underlying study indicate that that the high recognition rate is also independent of training level expressed by an anaerobic threshold speed (between 10.4 and 16.1 km/h) and running hours per week (ranging from one to six hours).

Even though a very high recognition rate was observed when compared to the same subject, it is still evident that the recognition percentages between different subjects were low, but not zero. Regarding the general curve structure of the detection corridor (Figure 7, blue and orange line), a high concordance over all subjects can be seen. The distinct differences, meaning the variation in the similarity rates, are, consequently, due to the individual variations of the general course. This means that a common pattern and subject specific variations exist more or less within the limits (Figure 7, yellow line), generated as a range based on all subjects’ data. The above described finding (see also Figure 7) is an indication for a general running pattern, which is inherent in all subjects representing universal running kinematics. Previous literature has mainly proposed two-part running cycles, namely stance and swing components, which can further be divided into sub-sections [26,27]. According to this statement, running seems to be a quite global movement. Only by closer inspection, and applying a computer algorithm, can the individual differences be uncovered as a runner’s unique gaitprint.

The current paper provides a basis to determine the characteristics of a well-developed running style. By applying the Attractor Method, it is possible to identify a person with a very high probability while only using their running motion. The next step is to determine, in which section of the general running cycle the greatest variation occurs. This would provide the basis for the identification between athletes. To what extent can these sections be subdivided into phases and can patterns be found that are related to highly successful athletes? Additionally, how can kinematic descriptions and physiological parameters, such as oxygen consumption at a given running speed, be connected? These are inspiring questions, the answers to which may be provided based on the present work. While these questions focus on a short-term time interval, sports science is generally also interested in considering these factors under the influence of fatigue in the context of endurance disciplines. In other words, how do the kinematics of human running change over a long-term effort, like a marathon race? This is not only of interest concerning a stable running performance and the associated remaining high recognition rate, but also because of the increasing risk of injury, which could possibly be detected at an early stage. In a study conducted by Nicol et al. [28], the kinematic parameters of eight marathon runners were examined in a running test with three different speeds before and after the marathon event by video recording. While there were no meaningful differences at the group level, individual differences from pre to post were found. This is confirmed by earlier works, like the one by Williams et al. [29], who also described the high individuality of kinematic parameters. These findings support the results of the present work. In the future it will have to be answered whether the recognition rate remains high, even with fatigue-induced individual kinematic changes.

Another highly useful application of the current findings could be in the recognition of athletes in the context of virtual racing series in e-sports, such as those practiced on ZWIFT [30] or the IRONMAN virtual club [31]. These online applications invite athletes to global running and cycling events where they can participate from their living room equipped with a treadmill and a stationary bike. All of the machines are virtually connected so that the athletes can compete against each other. Further, it is possible to win not only virtual but also material prizes, possibly slots for a real-life world championship [32,33]. This possibly leads to an increasing rate of fraud attempts such as having a performance by another athlete who is in a better shape compared to oneself. To avoid this way of cheating, the current approach could offer the possibility of obtaining a baseline gaitprint, which could be easily recognized when performing later.

One limitation of the current study might be that the different sessions run by the same athlete always had the same speed, which could have led to an easier recognition due to more consistent motion kinematics. It is known that higher speeds are related to changes in stride length kinematics [34,35], leading to a higher variation within the data and, consequently, to a decreased probability of recognition. Thus, future work is necessary in order to minimize the recognition rate to an equally low level for the described recognition method. This is essential when desiring to apply the attractor method recognition approach in everyday life, or physically active situations, where natural speed changes occur continuously.

## 5. Conclusions

In summary, the Attractor Method approach allows for highly sensitive discriminations between runners with different performance prerequisites. In addition, the results of the current study show a general running pattern, which is so distinguishable through the individual characteristics of the participants, that a recognition rate of over 99% can be achieved. Based on this knowledge, future work can now gain deeper insight into applications regarding running quality, fatigue, and recognition, as in, for example, an esports context.

## Figures and Tables

**Figure 1 sensors-20-03810-f001:**
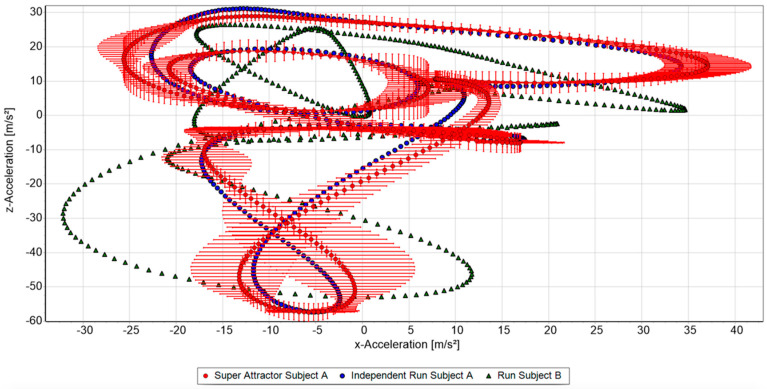
Two-dimensional depiction of the three-dimensional recognition horizon (red) compared to an attractor of the same (subject A, blue) and a different subject (B, green).

**Figure 2 sensors-20-03810-f002:**
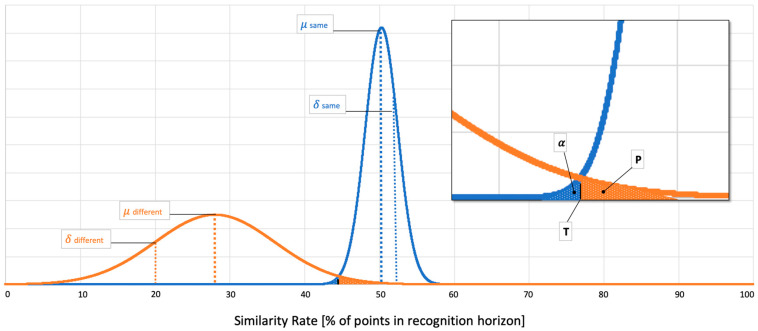
Example for two normally distributed probability curves (orange = between subject comparison; blue = within subject comparison; thin and thick dashed lines in orange and blue symbolize mean (μ) and standard deviation (*δ*); T is the border for rating different (left of T) and same (right of T); *α*-area = probability of not identifying a subject; P-area = probability of a false positive identification).

**Figure 3 sensors-20-03810-f003:**
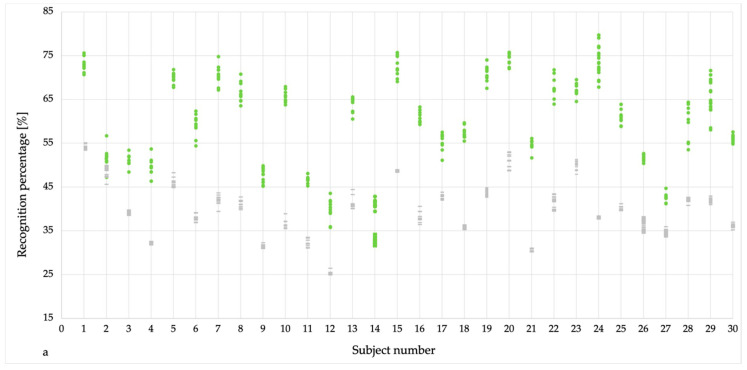
Recognition percentages presenting same-subject values (green dots in (**a**) & black crosses in (**b**)) and different-subject (worst-case) values (grey dashes in (**a**,**b**)).

**Figure 4 sensors-20-03810-f004:**
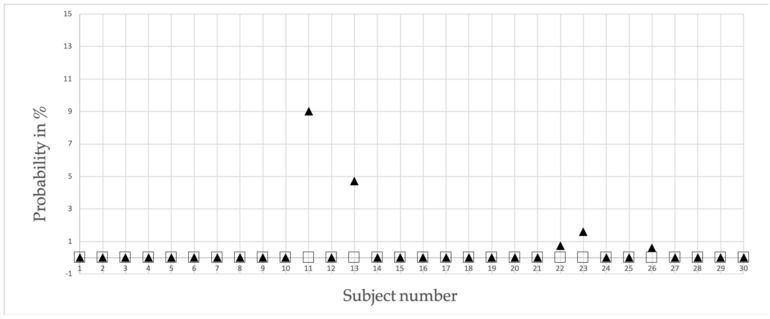
False positive detection probability chart displaying the individual percentages of all subjects. Squares = Super attractor versus the first run of the same person and filled triangles = Super attractor versus the second run of the same person. When probability is zero, squares are partly hidden by filled triangles.

**Figure 5 sensors-20-03810-f005:**
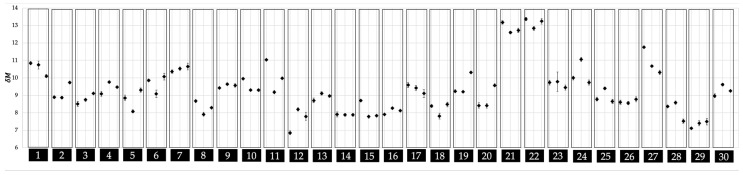
*δ*M mean of the morphing analysis with according standard deviations sorted by subjects.

**Figure 6 sensors-20-03810-f006:**
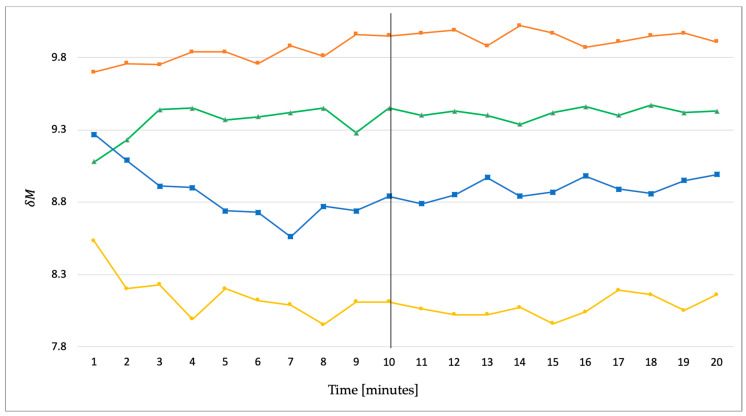
Four examples (subject 10 = orange, 9 = green, 2 = blue, 5 = yellow) showing different progressions of transient oscillations which have subsided after minute 10.

**Figure 7 sensors-20-03810-f007:**
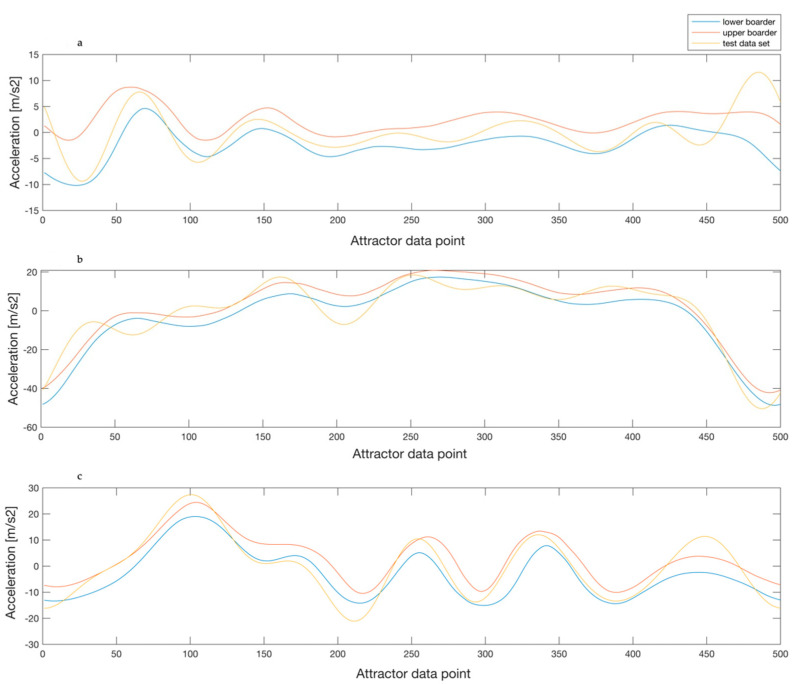
Display of the recognition horizon (blue and orange) with a data set from a differing subject over all three axes (**a** = *x*-axis, **b** = *y*-axis, **c** = *z*-axis). The recognition rate was 44%.

**Table 1 sensors-20-03810-t001:** Subject Overview.

	N	Age (yr)	Height (cm)	Weight (kg)	v_run_ (km/h)
**Female**	9	29 ± 5.2	169 ± 5.2	59.9 ± 5.6	12.4 ± 0.9
**Male**	21	36.8 ± 11.9	179 ± 6.4	73.9 ± 6.8	13.9 ± 1.4
**Overall**	30	34.7 ± 11.4	176 ± 7.6	69.7 ± 9.1	13.4 ± 1.5

**Table 2 sensors-20-03810-t002:** Overview of Characteristic Constants.

Constant	$T_{∥}$ Transient Effect’s Strength	$t_{T}$ Time for the Transient Effect Decreasing to T $\cdot e^{-1}$	$a_{0}$ Morphing’s Strength	$a_{1}$ Morphing’s Modulation Strength	$a_{2}$ Morphing’s Nonlinearity Strength
	−12.04–15.7	0.11–13.2	7–14	−0.39–39.6	0.09–1.69

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
