# Peer review of "The Gaitprint: Identifying Individuals by Their Running Style"

_sensors, 2020, doi:10.3390/s20143810_

Round 1
Reviewer 1 Report
This paper described an analysis method based on limit-cycle attractors to recognize subject’s patterns during running performances. The detection rate was 99 %, and it seems that running style is highly individual. The comments are as follows:
Major Comment,
- The advantage of MEMS sensor is that it can be easily used by the user, and can be measured in various environments unlike the video-based method. What is the accuracy of identification even in real situations, such as differences in the material of the ground (dirt or asphalt), and uphill or downhill? Can you achieve the same accuracy? Please discuss about these situation.
- Figure 6 is an interesting result, but since it is mentioned in the previous research [22], is this result just tracing the points of the previous research [22]? To clarify the contribution of this paper, the differences from the references should be clarified. What the author wanted to clarify in this paper is ambiguous to the reader because the method [22] is closely related to the method.
- Although the discussion describes the levels of exercisers such as beginners, this paper targets athletes. Please clarify whether the aim of this research is to target the running patterns of all people, regardless of the level of the exerciser, or to those who have experience.
Minor comment,
- LINE152, the font has changed. Authors should check the journal format.
- Figure 4, there are only four triangles. Was it buried in a square? Please improve.
- In Figure 7, why is test dataset above the lower boarder or upper boarder? Please describe the meaning of the lines correctly.
- Check journal format for code of ethics.
Reviewer 2 Report
Dear Authors,
In general, the submitted manuscript presents an interesting topic in the wearable devices context. The experiment is well-defined, especially since for each subject, multiple trials were performed (and on different days).
While the methodology presented is, in general, scientifically sound, I have a major concern regarding the compensation of differences in accelerometer positioning that was not taken into account. Namely,In Lines 225-226 it is stated that ‘For almost all cases there is a 
distinct gap visible between the same- and the different subject comparisons.’ However, this is by no means true - most of the subjects the second run values overlap with different subject values. Maybe I’m missing something but what can be concluded is that the horizon was calculated for run 1 for all subjects (hence the high similarity value) and that the method does not take into account slight differences in the positioning of the sensors form run one and two and hence misinterprets run 2 for the same subject for a run of a different subject.
Finally, while the manuscript itself is well-written, I did encountered some minor issues, mostly relating to the use of English language that have to be resolved. Since I most definitely did not encounter all of them, I suggest the manuscript is edited by a professional.
Minor concerns are, listed in order of appearance in the manuscript, as follows:
Abstract - Since ‘20’ in 20 minutes is written with numericals, I would advise using ‘30’ instead of thirty in thirty subjects.
Abstract Line 16 – there should not be a comma before ‘that’
Line 28 – 2max in VO2max 
should be subscript style
Line 30 - a comma should be used after ‘seventies’
Line 31 – ‘so called bio motion’ should be 
’so-called bio-motion’
Line 39 – ‘esports’ 
should be ‘e-sports’
Line 114 – Since the sentence starts with ‘They…’, ‘accelerometer’ should be ‘accelerometers’ and ‘Gyroscope’ should be ’gyroscopes’
Line 132 – ‘gyrometer’ should be replaced with gyroscope
Lines 146-147 - ‘a learning effect’ should be replaced with ‘the learning effect’
Line 183 – ‘a second step’ should be replaced with ‘the second step’
Line 202 – ‘Electronic noise caused by the MEMS sensors’ should be replaced with something like ‘MEMS sensors’ noise’

Line 103 – what is exactly meant with ‘suffered any present injury’? Is it ‘suffered any injury in the past’ or ‘don’t show any injury signs at this point’?
Line 115 - Was the sampling frequency consistently 500 Hz?
Lines 134-136 – the attractor method should be appropriately described (in short)
If the recognition horizon was calculated, why is Figure 1 taken from another source? It is also not explicitly stated that the horizon was calculated for both sensors' data (which I assume it was).
Round 2
Reviewer 1 Report
The paper has been well revised along with the comments provided by the reviewer. At this point, I would not have any more questions and review comments.Reviewer 2 Report
The authors adressed all of my concearns.